# REDEFINING GENERALIZATION IN VISUAL DOMAINS: A TWO-AXIS FRAMEWORK FOR FAKE IMAGE DETECTION WITH FUSIONDETECT

## ABSTRACT

The rapid development of generative models has made it increasingly crucial to develop detectors that can reliably detect synthetic images. Although most of the work has now focused on cross-generator generalization, we argue that this viewpoint is too limited. Detecting synthetic images involves another equally important challenge: generalization across visual domains. To bridge this gap, we present the **OmniGen Benchmark**. This comprehensive evaluation dataset incorporates 12 state-of-the-art generators, providing a more realistic way of evaluating detector performance under realistic conditions. In addition, we introduce a new method, `FusionDetect`, aimed at addressing both vectors of generalization. `FusionDetect` draws on the benefits of two frozen foundation models: CLIP & Dinov2. By deriving features from both complementary models, we develop a cohesive feature space that naturally adapts to changes in both the content and design of the generator. Our extensive experiments demonstrate that `FusionDetect` delivers not only a new state-of-the-art, which is **3.87%** more accurate than its closest competitor and **6.13%** more precise on average on established benchmarks, but also achieves a **4.48%** increase in accuracy on OmniGen, along with exceptional robustness to common image perturbations. We introduce not only a top-performing detector, but also a new benchmark and framework for furthering universal AI image detection.

## 1 INTRODUCTION

The field of artificial intelligence has entered an era of unprecedented creative capacity, primarily driven by the rapid maturation of text-to-image generative models Zhang et al. (2023). Recently, diffusion-based architectures such as Stable Diffusion Rombach et al. (2022), Midjourney MidJourney (2025), and Imagen Saharia et al. (2022) have achieved a level of photorealism and artistic flexibility that was once the domain of science fiction. These models have democratized content creation, empowering users to generate complex, high-fidelity images from simple textual descriptions. This technological leap has unlocked vast potential in domains ranging from digital art Saharia et al. (2022); Nichol et al. (2021) and entertainment Stark (2024) to product design Wang et al. (2025) and scientific visualization Thampanichwat et al. (2025). However, this accessibility is a double-edged sword. The same tools that foster creativity can be wielded for malicious purposes, including the generation of

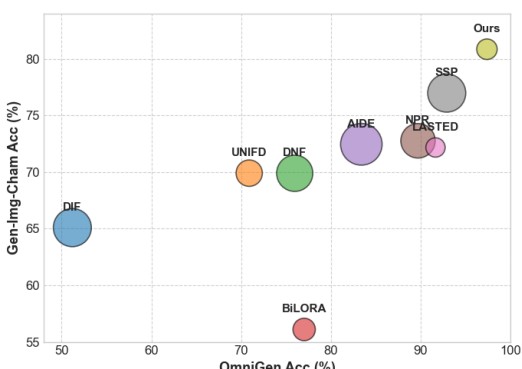

Figure 1: `FusionDetect` performance on OmniGen and established benchmarks from previous works Zhu et al. (2024); Boychev & Cholakov (2024); Yan et al. (2024) compared to other detectors. The size of the bubble indicates the standard deviation of accuracy between all generators in the dataset (smaller is better).

convincing disinformation, the creation of syn-
thetic media to erode public trust, and the violation of copyright and personal identity Xu et al.
(2023); Ren et al. (2024); Samrouth et al. (2024). Consequently, the development of robust, reli-
able, and universal methods for detecting AI-generated images has become a critical imperative for
ensuring the integrity of our digital ecosystem Mahara & Rishe (2025).

The academic pursuit of AI-generated image detection has evolved significantly, yet it faces persis-
tent challenges that limit its real-world applicability. Early research Wang et al. (2020); Zhang et al.
(2019); Qian et al. (2020) focused heavily on identifying artifacts from Generative Adversarial Net-
works (GANs) Goodfellow et al. (2014). Although fundamental, this focus is increasingly obsolete
due to the overwhelming dominance of diffusion models Ho et al. (2020); Song et al. (2021). These
models Ho et al. (2020); Dhariwal & Nichol (2021); Song et al. (2021); Rombach et al. (2022) are
the backbone of nearly all state-of-the-art (SOTA) commercial, open-source, and community-driven
projects. A modern, practical detector must therefore be also engineered for the unique and subtle
characteristics of this new paradigm.

More critically, we argue that the community's understanding of generalization is dangerously in-
complete as they focus on only one aspect of it. This typically involves training a detector on images
from a single generator and evaluating its ability to identify images from a variety of other gener-
ators Ojha et al. (2023); Wang et al. (2023); Yan et al. (2024). To rectify this, we formalize the
problem as a two-axis generalization challenge: a truly universal detector must demonstrate robust-
ness not only on the well-studied *cross-generator axis* (handling unseen generators) but also on
the often-neglected *cross-semantic axis* (handling unseen visual domains). As we will show, prior
works often fails on the second axis, rendering it unreliable for real-world Yan et al. (2024). This
semantic gap is not merely theoretical. As our t-SNE Maaten & Hinton (2008) projection in Fig-
ure 2 visualizes, popular datasets like GenImage Zhu et al. (2024), ImagiNet Boychev & Cholakov
(2024), and the challenging Chameleon Yan et al. (2024) form distinct, non-overlapping clusters in
our proposed `FusionDetect` embedding space. As shown, there is no to little overlap between
each dataset cluster. This demonstrates that a model trained exclusively on the feature distribution
of one dataset will fail to recognize the patterns of another, regardless of the generator used.

To solve this two-axis challenge, we propose
`FusionDetect`, a powerful fusion model
engineered for universal AI image detection.
We hypothesize that a truly robust and gen-
eralizable representation can only be created
by combining the complementary strengths of
large-scale, foundational models with orthog-
onal training objectives. Instead of hunt-
ing for a single, elusive universal artifact,
`FusionDetect` fuses deep features from two
distinct and powerful vision encoders: CLIP
Radford et al. (2021) for its unparalleled se-
mantic breadth and DINOv2 Oquab et al.
(2023) for its profound understanding of fine-
grained structure and texture.

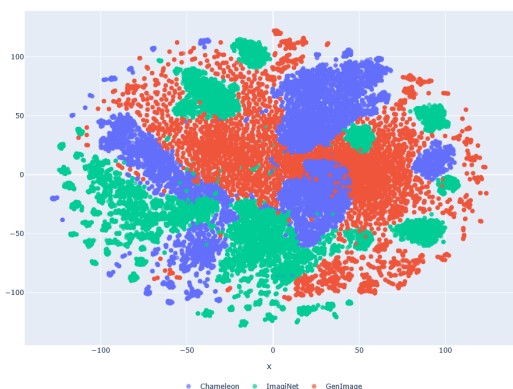

Figure 2: T-SNE Maaten & Hinton (2008) pro-
jection of GenImage Zhu et al. (2024), ImagiNet
Boychev & Cholakov (2024), and Chameleon Yan
et al. (2024) dataset.

To facilitate a more rigorous and realistic evalu-
ation of detector performance, we introduce the
**OmniGen** Benchmark, a new, open-source test
set designed to capture the modern generative
landscape. The OmniGen benchmark directly
addresses the weaknesses of prior benchmarks
by including images from 12 SOTA generators, such as closed-source models, the latest open-source
architectures, and popular community fine-tunes. By curating this benchmark with high semantic
variety, we provide a robust framework to validate a model's performance along both axes of gener-
alization, ensuring that our evaluations reflect a detector's true capabilities in real-world scenarios.
1

In summary, the primary contributions of this paper are fourfold:

1. We formalize the "two-axis generalization" problem in AI image detection, highlighting the critical need for models to generalize across both unseen generators and semantic domains.

2. We introduce `FusionDetect` as a strong proof-of-concept for this framework. It demonstrates that fusion of complementary foundational features can decisively outperform more complex architectures when evaluated under the two-axis setting.

3. We release the **OmniGen Benchmark**, the first test set explicitly designed to test two-axis generalization, featuring 12 diverse SOTA generators and high semantic variance.

4. We demonstrate through extensive experiments that `FusionDetect` establishes a new SOTA, achieving superior generalization and robustness to common image perturbations compared to existing methods.

## 2  RELATED WORK

The field of AI-generated image detection is in a constant race against generative technology. To provide context for our work, we'll first review the evolution of generative models, from older GANs Goodfellow et al. (2014); Karras (2017); Brock et al. (2018) to modern diffusion models Dhariwal & Nichol (2021); Rombach et al. (2022); Nichol et al. (2021). We'll then look at the detection methods, highlighting how each has responded to the shifting capabilities of generative architectures. Our review shows that existing detection methods have consistently lagged behind generative advancements, a critical gap that our work aims to close by addressing the "two-axis generalization" problem.

### 2.1  IMAGE GENERATION

The field of synthetic image generation has been reshaped over the last decade. It has transitioned from early breakthroughs with GANs Karras (2017); Karras et al. (2019); Brock et al. (2018) to the current dominance of Diffusion Models (DMs). The advent of Denoising Diffusion Probabilistic Models (DDPMs) Ho et al. (2020) marked a significant paradigm shift. Diffusion Models (DMs) and their subsequent variants have now surpassed GANs in terms of image quality, diversity, and text-to-image coherence Dhariwal & Nichol (2021). The initial wave of practical diffusion models was led by the Latent Diffusion Model (LDM) architecture Rombach et al. (2022), which underpins the widely popular Stable Diffusion series. These models made high-fidelity generation accessible to the public and became a foundational tool for both research and creative applications.

The pace of innovation has since accelerated, leading to a new generation of even more sophisticated architectures. Architectural upgrades in models like Stable Diffusion XL (SDXL) Podell et al. (2023), such as a larger UNet Ronneberger et al. (2015) and dual text-encoders, have led to significant improvements in image quality and prompt fidelity. The field continues to evolve rapidly with new open-source models like FLUX Labs et al. (2025), SD3.5 Esser et al. (2024), HiDream Cai et al. (2025), CogView4 Zheng et al. (2024), Kandinsky3 Vladimir et al. (2024), PixArt-$\delta$, alongside closed-source counterparts like Google's Imagen Saharia et al. (2022) and Midjourney MidJourney (2025) and also community finetuned models such as Juggernaut RunDiffusion (2025) and Dreamshaper Lykon (2025). This model shift from GANs to diffusions generates a new class of synthetic images with distinct statistical fingerprints that challenge existing detection methodologies, a primary focus of this work.

### 2.2  IMAGE DETECTION

Detection methodologies can be broadly categorized into two main paradigms: those that seek to identify specific, inherent artifacts of the generation process, and those that leverage the general-purpose feature representations of large pretrained foundational models.

#### ARTIFACT-BASED DETECTION

This paradigm is founded on the principle that the synthetic generation process, regardless of its sophistication, leaves behind subtle, machine-detectable traces or "fingerprints" Sinitsa & Fried (2024). Researchers have pursued these artifacts across various domains. A significant body of

work targets universal image properties, analyzing inconsistencies in the frequency domain (Frank et al. (2020); Tan et al. (2024); Sinitsa & Fried (2024); Qian et al. (2020)), exploring local texture and patch-level correlations (Zhong et al. (2023); Tan et al. (2023); Chen et al. (2024b)), or extracting unique residual noise patterns left by the generation process (Zhang & Xu (2023); Liu et al. (2022)). More recently, a modern class of artifact-based detectors leverages the internal mechanics of the diffusion process itself as a forensic tool. This approach is broadly divided into error-based and non-error-based methods. Error-based detectors operate on the principle that diffusion models reconstruct their own outputs with lower error than real images, using this discrepancy in pixel space (Wang et al. (2023); Ma et al. (2023)), in latent space (Ricker et al. (2024)), or as a guiding feature (Luo et al. (2024)). In contrast, non-error-based methods use the diffusion pipeline in other ways, such as to generate hard negative training samples (Chen et al. (2024a)), to extract internal representations like noise maps as features (Cazenavette et al. (2024)), or to distill a slow, error-based model into a faster one (Lim et al. (2024)). A detailed overview of these detection paradigms is provided in Appendix 7.3.

Despite their successes, our experiments indicate that artifact-based detection methods face significant limitations. First, their performance is often brittle, demonstrating poor cross-generator generalization. As generative models evolve, the specific artifacts these methods rely on change, making the detectors quickly outdated. Second, they are highly sensitive to common image perturbations, like compression, which can easily destroy the subtle fingerprints they detect.

### PRETRAINED FEATURE-BASED DETECTION

A more recent and increasingly dominant paradigm moves away from specialized artifact detection and instead leverages the rich feature spaces of large-scale, pretrained foundational models Yan et al. (2024); Ojha et al. (2023); Keita et al. (2025). The core idea is that these models, having been trained on web-scale data, have learned robust and generalizable representations of the visual world. A pioneering work in this area is `UniFD` Ojha et al. (2023), which demonstrated that a simple linear classifier trained on CLIP Radford et al. (2021) features can achieve impressive generalization across unseen generators. This highlighted the power of semantic features for the detection task. Other works have explored this vision-language connection further; for example, `Bi-LoRa` Keita et al. (2025) reframes the detection problem as a visual question-answering or captioning task. Methods like `LASTED` Wu et al. (2023) also leverage language-guided contrastive learning. `AIDE` Yan et al. (2024), proposed a hybrid model that combined semantic features from a pretrained CLIP model with specialized, hand-crafted modules (DCT Ahmed et al. (2006) and SRM Fridrich & Kodovsky (2012) filters) to capture low-level texture statistics.

The success of sophisticated hybrid approaches like AIDE Yan et al. (2024), raises a critical question: is it necessary to design hand-crafted modules for low-level features, or can a more effective and less complex solution be found by fusing the features of two distinct, general-purpose foundational models? To answer this question we proposed `FusionDetect` that utilized feature fusion of foundation models and experiment on the impact of such approach.

## 3 METHODOLOGY

This section details `FusionDetect`, a model explicitly designed to solve the two-axis generalization problem. We formally define this as training a detector $D$ on a distribution of generators $G_{train}$ and semantic domains $S_{train}$ that must generalize to a test set drawn from $G_{test}$ and $S_{test}$, where $G_{train} \cap G_{test} = \emptyset$ and $S_{train} \cap S_{test} = \emptyset$. `FusionDetect` addresses this by creating a hybrid feature space engineered to be a strong baseline detector invariant to shifts in both $G$ and $S$.

### FUSIONDETECT

The feature extraction backbone of `FusionDetect` consists of two distinct, powerful vision encoders. A key design choice is that both of these pretrained backbones remain frozen during training. This helps with computation efficiency and faster training time as well as preventing the model from overfitting to the training data and preserving the highly generalizable, world-knowledge features learned by these models during their original large-scale pretraining.

The two branches of our feature extractor are as follows:

1. **Semantic Feature Encoder (CLIP):** We employ a CLIP vision encoder to capture high-level semantic, contextual, and object-level information. Its rich understanding, derived from large-scale image-text pretraining, is crucial for achieving cross-semantic generalization. Given an input image $I$, the CLIP encoder $E_{CLIP}$ produces a semantic feature vector.

2. **Structural Feature Encoder (DINOv2):** We use a DINOv2 vision transformer to capture fine-grained structural and textural details. As a self-supervised model, it is highly sensitive to the low-level patterns and artifacts that betray a synthetic origin, which is vital for achieving cross-generator generalization. The DINOv2 encoder $E_{DINO}$ processes the same input image $I$ to produce a structural feature vector.

These two feature vectors are then fused via concatenation to form a comprehensive hybrid feature vector, $z_f$, assuming $d_{clip}$ and $d_{dino}$ are the image encoders output dimensions:

$$z_f = [E_{CLIP}(I) \in \mathbb{R}^{d_{clip}} \parallel E_{DINO}(I) \in \mathbb{R}^{d_{dino}}] \in \mathbb{R}^{d_{clip}+d_{dino}} \tag{1}$$

$z_f$, is then processed by the only trainable component of our model: a lightweight Multi-Layer Perceptron (MLP) classifier head, denoted by the function $f_\theta$ with parameters $\theta$. The model is trained end-to-end to minimize the binary cross-entropy (BCE) loss $L(\theta)$ over a batch of $N$ images, defined as:

$$L(\theta) = -\frac{1}{N} \sum_{i=1}^{N} [y_i \log(p_i) + (1 - y_i) \log(1 - p_i)] \tag{2}$$

where $y_i \in \{0, 1\}$ is the ground-truth label and $p_i = \sigma(f_\theta(z_{f,i}))$ is the predicted probability from the sigmoid function $\sigma$.

## 4 EXPERIMENTS

To empirically validate the effectiveness of our proposed `FusionDetect` model, we conduct a series of comprehensive experiments. Our evaluation is designed to rigorously test performance along the two-axis generalization problem, assess robustness to real-world image perturbations, and dissect the model's architecture to understand the contribution of its core components.

### 4.1 EXPERIMENTAL SETUP

**Implementation Details:** The `FusionDetect` model was trained for an efficient 10 epochs using an AdamW optimizer on a single NVIDIA RTX 3090 GPU. The final architecture consists of frozen *CLIP-ViT-L14* and *DINOv2-L14* backbones and a 4-layer MLP classifier head. To enhance robustness, we applied random JPEG compression and Gaussian blur to 10% of the images during training.

**Baselines for Comparison:** We compare `FusionDetect` against a comprehensive suite of recent detectors which their code and pretrained weights were publicly accessible: `DIF` Sinitsa & Fried (2024), `UNIFD` Ojha et al. (2023), `DNF` Zhang & Xu (2023), `LASTED` Wu et al. (2023), `BiLORA` Keita et al. (2025), `AIDE` Yan et al. (2024), `SSP`, and `NPR`. To ensure a thorough and fair evaluation, these models are tested in two settings where applicable: using their original, off-the-shelf pretrained weights, and after being retrained from scratch on our custom dataset.

**Training Dataset:** To directly address the semantic generalization gap, we curated a custom, balanced training set of 60,000 images (30k real, 30k fake). This dataset was constructed to maximize categorical and stylistic diversity by combining three distinct sources: samples derived from the large-scale ImagiNet Boychev & Cholakov (2024) and GenImage Zhu et al. (2024) benchmarks, and a challenging set of images generated using prompts derived from the hyper-realistic Chameleon dataset Yan et al. (2024). To follow the training scheme of previous work for cross-generator generalization, only images generated by SD1.4 and SD2.1 Rombach et al. (2022) were used in the train dataset and tested on others generators and datasets.

**Evaluation Metrics:** Similar to previous work, we report performance using Accuracy (Acc) and Average Precision (AP) Yan et al. (2024); Wu et al. (2023); Tan et al. (2024); Ojha et al. (2023). To

Table 1: Performance comparison on established benchmarks. Models marked with * were evaluated using their official pretrained weights. All other baselines were retrained on our custom training set. Results are in the format: Acc / AP (%). Best overall performance is **bold**, second best is underlined.

| Detector | GenImage | ImagiNet | Chameleon | STD | Mean |
|---|---|---|---|---|---|
| BiLoRA* | 61.20 / 61.13 | 56.56 / 53.61 | 50.54 / 42.21 | 5.34 / 9.53 | 56.10 / 52.32 |
| DIF* | 91.80 / 65.20 | 50.20 / **99.60** | 53.30 / **87.10** | 23.17 / 17.41 | 65.10 / 81.97 |
| UNIFD* | 70.49 / 88.93 | 76.16 / 85.67 | 50.76 / 54.49 | 13.33 / 19.01 | 65.80 / 76.36 |
| NPR* | 75.70 / 81.70 | 74.60 / 74.50 | 54.20 / 37.89 | 12.11 / 23.49 | 68.17 / 64.70 |
| DNF | 77.90 / **99.71** | 74.42 / 91.79 | 57.37 / 35.30 | 10.99 / 35.12 | 69.90 / 75.60 |
| UNIFD | 73.42 / 79.02 | 71.74 / 79.01 | 64.56 / 64.58 | 4.71 / 8.33 | 69.91 / 74.20 |
| LASTED | 73.14 / 61.20 | 74.10 / 59.98 | 69.31 / 62.33 | **2.53** / 1.18 | 72.18 / 61.17 |
| AIDE | 88.51 / 97.09 | 69.23 / 85.46 | 59.63 / 67.49 | 14.71 / 14.91 | 72.46 / 83.35 |
| LASTED* | 93.61 / 76.74 | 62.74 / 59.88 | 61.42 / 55.27 | 18.22 / 11.30 | 72.59 / 63.96 |
| NPR | 86.50 / 94.10 | 72.70 / 87.60 | 59.10 / 61.50 | 13.70 / 17.25 | 72.77 / 81.07 |
| AIDE* | 87.11 / 98.06 | 72.65 / 83.73 | 61.82 / 63.08 | 12.69 / 17.58 | 73.86 / 81.62 |
| SSP* | **93.59** / 99.10 | 75.61 / 82.81 | 58.64 / 64.24 | 17.48 / 17.44 | 75.95 / 82.05 |
| SSP | 93.34 / 96.86 | 77.99 / 80.19 | 59.64 / 66.77 | 16.87 / 15.07 | 76.99 / 81.27 |
| **FusionDetect** | 83.03 / 91.28 | **83.23** / 90.91 | **76.32** / 80.02 | 3.93 / **6.41** | **80.86 / 87.40** |

specifically measure generalization, we compute the both the Mean and Standard Deviation (STD) of these metrics across diverse benchmarks. A lower STD is a critical indicator of a robust detector, as it signifies consistent performance across different semantic domains. Accuracy of each class (real/fake) has also been reported in Appendix 7.4.

## 4.2 COMPARATIVE ANALYSIS ON ESTABLISHED BENCHMARKS

To validate the generalization capabilities of `FusionDetect` and provide a comparison point to prior work, we evaluated it on a collection of diverse and established test sets. The test set contains 8000, 10000, and 2595 images from GenImage Zhu et al. (2024), ImagiNet Boychev & Cholakov (2024) and Chameleon Yan et al. (2024) datasets respectively, each containing equal number of real and synthetic images. To ensure a fair and comprehensive evaluation, the test set is composed of an equal number of images sampled from every available generator within each source dataset. A detailed overview can be found in Appendix 7.2.

As shown in Table 1, `FusionDetect` achieves the best overall performance, attaining the highest mean accuracy and average precision, and crucially, the lowest standard deviation. It surpasses the closest competitor by $3.87\%$ in accuracy, $6.13\%$ in average precision. An important note is that although other detectors perform better on GenImage Zhu et al. (2024), but on ImagiNet Boychev & Cholakov (2024) and the difficult chameleon Yan et al. (2024) dataset our detector outperforms others by almost $10\%$ on average which indicates that previous models were specifically designed to perform well on the GenImage benchmark since it was seen as the standard benchmark for synthetic image detection; and not as a universal detector. Acheiving low STD and high mean accuracy on three different datasets indicates the domain generalization capabilities of `FusionDetect`.

## 4.3 THE OMNIGEN BENCHMARK

A core contribution of our work is the creation of a new, open-source benchmark designed to reflect the practical challenges of AI image detection. Our motivation was to address the shortcomings of existing benchmarks, which often lack semantic diversity and lag behind the rapid pace of generator development. The OmniGen benchmark was designed to be more practical and challenging by focusing on the latest SOTA generators, including both closed-source APIs and popular fine-tuned community models.

**Generator Selection:** The benchmark contains 11,550 fake images from a curated list of 12 relevant and powerful generative models, categorized as follows:

Table 2: Comparison of detectors on our proposed OmniGen test set on Accuracy (%). Models marked with * were evaluated using their official pretrained weights. Best results are in **bold** and second best is underlined.

| Generator | DIF* | UNIFD* | UNIFD | DNF | LASTED* | BILORA* | AIDE* | AIDE | SSP* | NPR | LASTED | SSP | Ours |
|---|---|---|---|---|---|---|---|---|---|---|---|---|---|
| GPT-4o | 52.8 | 71.5 | 73.9 | 82.9 | 82.1 | 79.6 | 87.6 | 82.9 | 97.2 | **99.4** | 85.7 | 98.3 | 97.3 |
| Imagen 4 | 50.3 | 49.6 | 62.0 | 72.8 | 73.7 | 79.4 | 49.2 | 51.2 | 56.8 | 88.6 | 96.7 | 74.9 | **97.5** |
| Imagen 4 Ultra | 50.2 | 50.0 | 61.9 | 72.7 | 72.7 | 80.0 | 49.8 | 51.2 | 57.3 | 87.3 | **96.8** | 75.6 | 96.4 |
| FLUX 1 dev | 57.6 | 49.3 | 70.5 | 86.7 | 73.0 | 80.5 | 83.4 | 83.7 | 97.8 | 94.2 | 88.3 | 96.7 | **98.5** |
| Kandinsky 3 | 50.2 | 61.4 | 67.7 | 88.2 | 84.0 | 70.2 | 81.4 | 90.1 | 97.8 | 92.8 | 91.2 | 97.7 | **99.3** |
| PixArt-$\delta$ | 51.3 | 60.1 | 77.7 | 88.7 | 82.7 | 74.7 | 85.4 | 88.3 | 96.6 | 84.3 | 90.2 | 94.8 | **99.0** |
| Juggernaut v11 | 50.1 | 52.9 | 76.0 | 59.8 | 91.3 | 76.8 | 93.2 | 94.0 | 96.6 | 91.3 | 94.0 | 97.2 | **99.2** |
| Dreamshaper | 50.2 | 54.4 | 77.3 | 65.2 | 85.4 | 74.8 | 95.6 | 95.2 | 99.4 | 83.1 | 96.9 | **99.9** | 98.4 |
| CogView4-6B | 50.0 | 49.9 | 80.5 | 61.5 | 57.5 | 74.1 | 94.6 | 94.8 | 99.1 | 97.6 | 90.4 | 92.9 | **99.6** |
| HiDream-I1 | 50.4 | 49.3 | 68.5 | 89.6 | 75.7 | 74.7 | 61.7 | 90.2 | 93.7 | 87.7 | 91.6 | 96.9 | **97.9** |
| SD3.5-medium | 50.1 | 56.6 | 76.6 | 72.7 | 74.6 | 79.9 | 80.0 | 87.5 | 94.1 | 95.5 | 86.5 | 92.2 | **98.2** |
| MidJourney v7 | 51.2 | 68.0 | 58.2 | 71.0 | 60.2 | 79.7 | 76.5 | 91.8 | 86.6 | 74.6 | 91.6 | **98.0** | 87.5 |
| **STD** | **2.17** | 7.68 | 7.30 | 11.94 | 9.95 | 3.30 | 16.23 | 15.55 | 15.51 | 6.97 | 3.84 | 8.54 | 3.26 |
| **Mean** | 51.20 | 56.06 | 70.90 | 75.97 | 76.06 | 77.02 | 78.18 | 83.39 | 89.40 | 89.70 | 91.65 | 92.90 | **97.38** |

- *Closed-Source*: GPT-4o Hurst et al. (2024), Imagen 4 Saharia et al. (2022), Imagen 4 ultra Saharia et al. (2022), MidJourney v7 MidJourney (2025).

- *Open-Source*: FLUX 1 Labs et al. (2025), Kandinsky 3 Vladimir et al. (2024), PixArt-$\delta$ Chen et al. (2024c), SD3.5-medium Esser et al. (2024), HiDream-I1 Cai et al. (2025), CogView4-6B Zheng et al. (2024).

- *Community Fine-tuned:* Juggernaut RunDiffusion (2025), Dreamshaper Lykon (2025).

**Benchmark Curation:** To ensure high semantic diversity and prevent model overfitting to specific concepts, the synthetic images were generated using a structured, randomized prompt template:

*"A richly detailed, high-resolution and photorealistic image depicting: {subject} during the {time}. The scene includes {setting}, {visual}, and lifelike rendering. The image style resembles {style}. Use {light}."*

Each bracketed variable was populated from a large pool of options (e.g., over 400 different subjects), resulting in highly unique prompts for each image. For each of the 12 generators, we generated 1000 images with $1024 \times 1024$ resolution which were evaluated against a set of 1000 real images from Unsplash uns. A detailed overview and examples generated can be found in Appendix 7.1.

Our secondary evaluation is conducted on the new OmniGen benchmark, which is designed to test detectors against the modern, real-world generative landscape. The results shown in Table 2 demonstrate the superior performance of `FusionDetect`, specifically in accuracy where we see $+4.48\%$ increase in performance. Among the detectors, `FusionDetect` is the top performer, achieving the highest mean accuracy of $97.38\%$ and a remarkably low standard deviation of 3.26. This indicates its consistent ability to generalize across a wide variety of SOTA generators, from closed-source APIs to open-source and community. AP and class accuracies are reported in 7.4.

## 4.4 ROBUSTNESS TO COMMON PERTURBATIONS

A critical attribute of a practical detector is its resilience to image degradations commonly encountered online. We subjected the detectors to two stress tests: JPEG compression and Gaussian blur. The results, shown in Table 3, highlight a key weakness in many artifact-based detectors. Models that rely on high-frequency spatial artifacts, such as `SSP` Chen et al. (2024b), `NPR` Tan et al. (2024) and `DNF` Zhang & Xu (2023), fail drastically under both compression and blur, despite their high scores on clean images. In contrast, `FusionDetect`'s performance barely sees any degradation. This remarkable stability confirms that our model's decisions are based on more fundamental, robust features rather than fragile, easily-disrupted artifacts, making it far more suitable for real-world deployment.

Table 3: Robustness analysis under common image perturbations. All models are subjected to varying levels of JPEG compression and Gaussian blur. Results are reported as Acc / AP (%).

| Detector | No Degradation | JPEG Compression | | | Gaussian Blur | | |
|---|---|---|---|---|---|---|---|
| | | QF=95 | QF=75 | QF=50 | $\sigma = 1.0$ | $\sigma = 2.0$ | $\sigma = 3.0$ |
| DNF | 71.61 / 89.12 | 59.83 / 72.44 | 59.80 / 70.77 | 50.89 / 68.18 | 50.89 / 44.23 | 50.88 / 37.10 | 41.96 / 40.96 |
| UNIFD | 63.38 / 72.54 | 61.65 / 70.18 | 61.91 / 70.74 | 61.38 / 70.20 | 59.85 / 69.26 | 59.11 / 68.02 | 58.38 / 66.02 |
| LASTED | 70.35 / 57.58 | 69.43 / 55.94 | 69.83 / 57.53 | 69.14 / 55.28 | 69.34 / 57.68 | 69.12 / 57.19 | 69.82 / 57.22 |
| AIDE | 74.49 / 86.88 | 67.50 / 79.20 | 66.22 / 74.71 | 64.77 / 70.92 | 68.32 / 76.47 | 67.82 / 76.81 | 66.66 / 75.64 |
| NPR | 74.33 / 85.50 | 52.30 / 53.50 | 51.50 / 52.50 | 51.30 / 51.90 | 66.30 / 72.80 | 56.20 / 60.20 | 54.50 / 56.80 |
| SSP | 76.86 / 86.72 | 52.77 / 63.11 | 51.32 / 55.66 | 51.05 / 54.83 | 57.48 / 61.74 | 54.49 / 58.64 | 53.27 / 57.47 |
| **FusionDetect** | **80.91 / 90.43** | **80.94 / 91.04** | **81.03 / 91.10** | **80.45 / 91.08** | **80.92 / 92.64** | **79.32 / 92.34** | **78.87 / 92.00** |

Table 4: Ablation and sensitivity analysis of `FusionDetect`. (a) Ablation on core components. (b) Sensitivity to different backbone and classifier architectures.

(a) Component Ablation Study.

| CLIP | DINOv2 | FeatUp | Acc / AP (%) |
|---|---|---|---|
| ✓ | ✗ | ✓ | 62.83 / 70.47 |
| ✗ | ✓ | ✓ | 60.69 / 68.38 |
| ✓ | ✓ | ✓ | 66.18 / 72.49 |
| ✓ | ✗ | ✗ | 77.81 / 85.77 |
| ✗ | ✓ | ✗ | 75.42 / 86.11 |
| ✓ | ✓ | ✗ | **80.92 / 92.86** |

(b) Sensitivity Analysis.

| Variable | Variants | Acc / AP (%) |
|---|---|---|
| CLIP *(DINO: ViT-L14)* *(Classifier: 4 layer)* | - | 78.81 / 86.77 |
| | ViT-H14 quickgelu | 79.37 / 90.48 |
| | ViT-L14 quickgelu | 80.14 / 91.36 |
| | **ViT-L14** | **80.92 / 92.86** |
| DINOv2 *(CLIP: ViT-L14)* *(Classifier: 4 layer)* | - | 75.98 / 84.15 |
| | ViT-S14 | 79.21 / 90.17 |
| | ViT-B14 | 79.51 / 91.12 |
| | **ViT-L14** | **80.92 / 92.86** |
| Classifier *(CLIP: ViT-L14)* *(DINO: ViT-L14)* | 1 layer | 80.26 / 89.55 |
| | 2 layers | 80.33 / 89.47 |
| | 3 layers | **81.14** / 92.58 |
| | **4 layers** | 80.92 / **92.86** |
| | 5 layers | 80.75 / 92.92 |

## 4.5 ABLATION AND SENSITIVITY ANALYSIS

We evaluated the performance of the individual components of our model. As shown in Table 4a, while only using the CLIP Radford et al. (2021) model as feature extractor performs well, consistent with findings from prior work, and only using the DINOv2 Oquab et al. (2023) model is also effective, their fusion in `FusionDetect` yields the best results. This confirms our core hypothesis that the two models provide complementary features. The t-SNE visualization of these three embeddings is shown in Figure 3 which indicates that that CLIP+DINO embedding can easily separate not only real and fake images but also their underlying dataset. We also explored incorporating feature up-scaling via FeatUp Fu et al. (2024), but this did not improve performance, suggesting that the raw, powerful features from the foundational backbones are more discriminative for this task.

Moreover, we analyzed the impact of different backbones and classifier depths on performance and the results are shown in Table 4b. The choice of *ViT-L/14* for both backbones was made for consistency with prior work and to leverage the power of large-scale architectures. The results show that a relatively simple 4-layer MLP classifier is sufficient to achieve SOTA performance. This indicates that the true power of `FusionDetect` lies in its robust feature extractor, which is so effective that it does not require a complex classifier to learn a decision boundary.

## 5 DISCUSSION

The empirical findings presented here strongly substantiate our central thesis regarding the "two-axis generalization" issue. `FusionDetect` exhibits remarkable stability across the cross-semantic axis, as evidenced by its minimal standard deviation on benchmarks such as GenImage Zhu et al. (2024), ImagiNet Boychev & Cholakov (2024), and Chameleon Zhong et al. (2023). This consistency indicates that its performance is not restricted by the underlying visual domain. Additionally, `FusionDetect` maintains impressive accuracy on the OmniGen benchmark which can be consid-

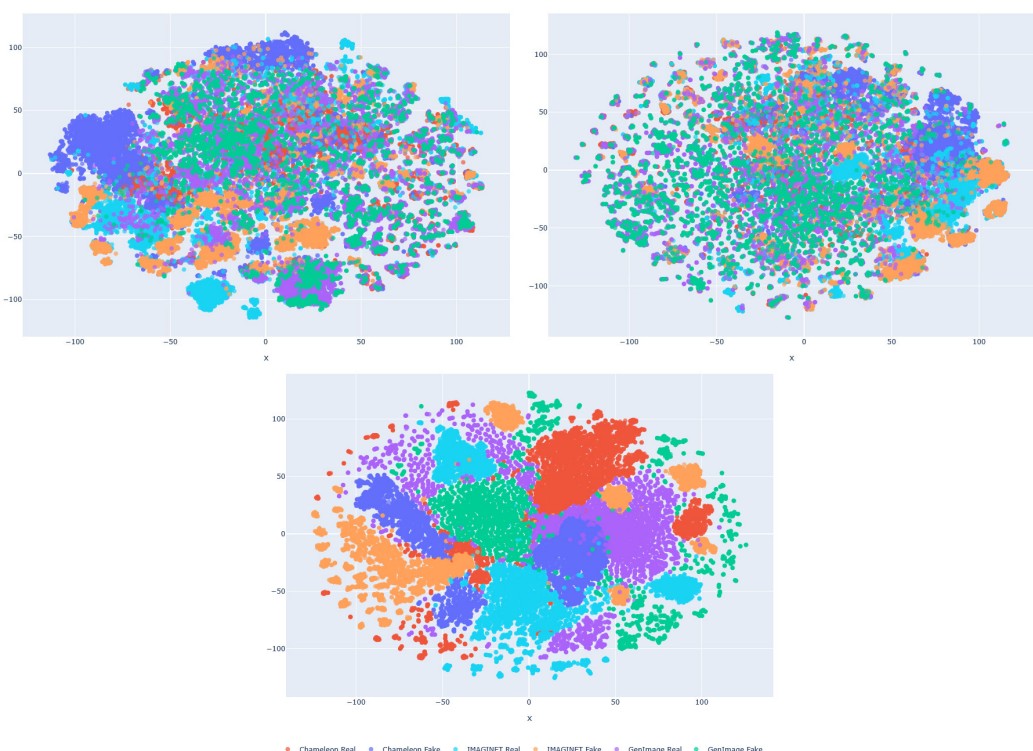

Figure 3: T-SNE Maaten & Hinton (2008) projection of GenImage Zhu et al. (2024), ImagiNet Boychev & Cholakov (2024), and Chameleon Yan et al. (2024) dataset. The CLIP+DINO (bottom) encoder successfully separates real and fake classes for each dataset unlike the other two options. (Top left: CLIP, Top right: DINOv2)

ered out-of-distribution both semantically and also unseen generators used, affirming its robustness across the two axes. Our proposed benchmark itself plays a pivotal role in this analysis. By incorporating both cutting-edge closed-source models and a variety of community fine-tuned models, it reveals the limitations of detectors that only excel on outdated test sets. These results underscore the necessity of benchmarks that capture both semantic and generator diversity to meaningfully assess a detector's real-world effectiveness.

## 6 CONCLUSION

In this paper, we redefined the task of AI-generated image detection by formalizing the "two-axis generalization" task that warrants robustness to both previously unseen generators and different semantic domains. To tackle the two-axis generalization task, we presented the **OmniGen Benchmark**, a new challenging test set consisting of 12 SOTA generators, and the `FusionDetect`, a robust detector that solves the two-axis problem by learning representations in a fusion model composed of complementary features extracted from foundational models. We show, empirically, that `FusionDetect` sets a state-of-the-art within generalization and robustness stages that indicate intelligently fusing complementing features extracted from foundational models is a better paradigm than building from scratch with special architectures. More generally, the two-axis framework offers a valuable method for evaluating model robustness that is broadly applicable, and we hope our contributions lay the groundwork for the next generation of universal fake media detectors.

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

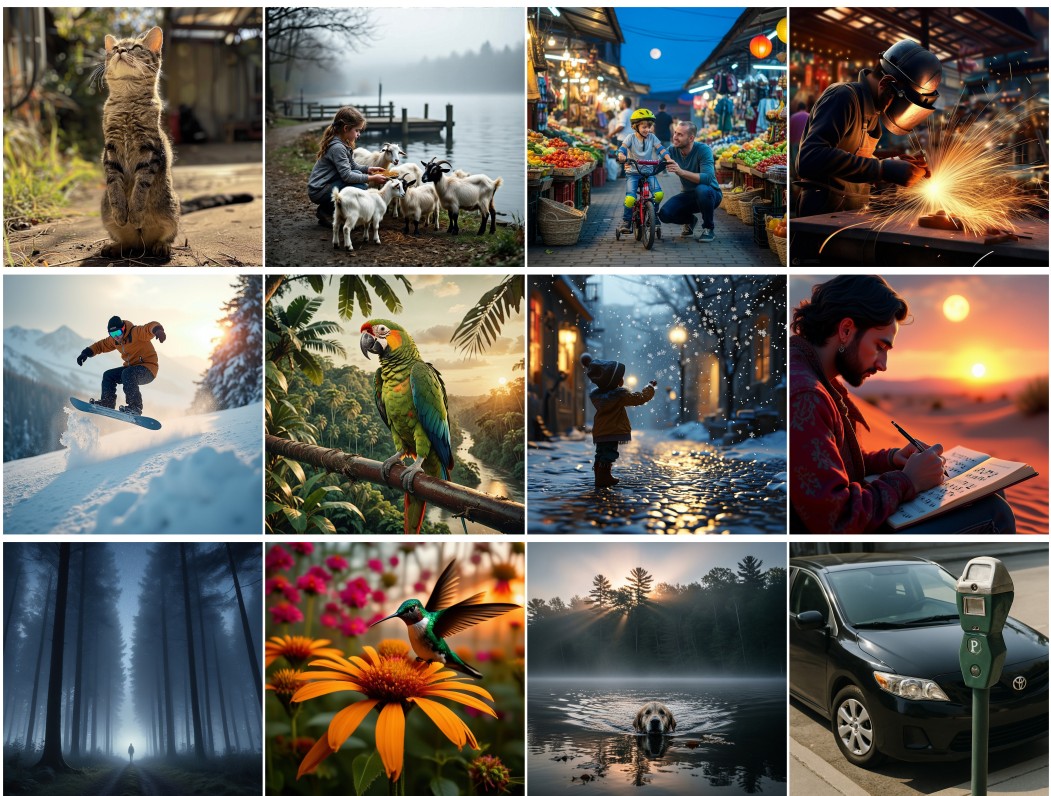

Figure 4: OmniGen Benchmark Images. **Top row:** Midjourney v7 MidJourney (2025), HiDream Cai et al. (2025), Imagine 4 Saharia et al. (2022), Kandinsky 3 Vladimir et al. (2024); **Middle row:** Flux 1 Labs et al. (2025), Dreamshaper Lykon (2025), Pixart-$\delta$ Chen et al. (2024c), Cogview 4 Zheng et al. (2024); **Bottom row:** Juggernaut RunDiffusion (2025), SD3.5 Esser et al. (2024), Imagen 4 ultra Saharia et al. (2022), GPT4o Hurst et al. (2024).

# 7 APPENDIX

## 7.1 DETAILED OVERVIEW OF OMNIGEN BENCHMARK

This appendix provides supplementary details for the OmniGen Benchmark introduced in Section 3. To offer a comprehensive overview of its composition, Table 5 lists all 12 state-of-the-art generators used in its creation, along with their respective image counts, resolutions, and sourcing methods. Furthermore, Figure 4 presents a selection of example images generated by these models, visually demonstrating the high degree of realism and semantic diversity that makes the OmniGen benchmark a challenging and realistic testbed for modern AI image detectors.

## 7.2 DETAILED OVERVIEW OF ESTABLISHED DATASETS

We provide additional information regarding the test sets used for GenImage Zhu et al. (2024), ImagiNet Boychev & Cholakov (2024), and Chameleon Yan et al. (2024) in Section 4. It includes generators used, number of images, image resolution, semantic categories, and the source for real images utilized in these datasets which is shown in Tables 6, 7, and 8. Note that the numbers reported in these tables indicate the count of fake images only.

Table 5: Composition of the OmniGen Benchmark generators containing 11550 synthetic images in total.

| Generator | Number | Resolution | Gather Method | Real Source |
|---|---|---|---|---|
| GPT-4o Hurst et al. (2024) | 550 | $1024 \times 1024$ | Public Dataset | |
| MidJourney v7 MidJourney (2025) | 1000 | $1024 \times 1024$ | Public Dataset | |
| Imagen 4 Saharia et al. (2022) | 1000 | $1024 \times 1024$ | API | |
| Imagen 4 Ultra Saharia et al. (2022) | 1000 | $1024 \times 1024$ | API | |
| FLUX 1 Labs et al. (2025) | 1000 | $1024 \times 1024$ | Local Model | |
| Kandinsky 3 Vladimir et al. (2024) | 1000 | $1024 \times 1024$ | Local Model | Unsplash |
| PixArt-$\delta$ Chen et al. (2024c) | 1000 | $1024 \times 1024$ | Local Model | |
| Juggernaut v11 RunDiffusion (2025) | 1000 | $1024 \times 1024$ | Local Model | |
| Dreamshaper Lykon (2025) | 1000 | $1024 \times 1024$ | Local Model | |
| CogView4-6B Zheng et al. (2024) | 1000 | $1024 \times 1024$ | Local Model | |
| HiDream-I1 Cai et al. (2025) | 1000 | $1024 \times 1024$ | Local Model | |
| SD3.5-medium Esser et al. (2024) | 1000 | $1024 \times 1024$ | Local Model | |

Table 6: Composition of the GenImage Zhu et al. (2024) evaluation set used. The test set contains 4000 fake and 4000 real images.

| Generator | Number | Resolution | Real Source |
|---|---|---|---|
| BigGAN Brock et al. (2018) | 500 | $512 \times 512$ | ImageNet Deng et al. (2009) |
| VQDM Gu et al. (2022b) | 500 | $256 \times 256$ | ImageNet Deng et al. (2009) |
| SDv1.4 Rombach et al. (2022) | 500 | $512 \times 512$ | ImageNet Deng et al. (2009) |
| SDv1.5 Rombach et al. (2022) | 500 | $512 \times 512$ | ImageNet Deng et al. (2009) |
| Wukong Gu et al. (2022a) | 500 | $512 \times 512$ | ImageNet Deng et al. (2009) |
| ADM Dhariwal & Nichol (2021) | 500 | $512 \times 512$ | ImageNet Deng et al. (2009) |
| Glide Nichol et al. (2021) | 500 | $512 \times 512$ | ImageNet Deng et al. (2009) |
| MidJourney v5 MidJourney (2025) | 500 | $1024 \times 1024$ | ImageNet Deng et al. (2009) |

Table 7: Composition of the ImagiNet Boychev & Cholakov (2024) evaluation set used. The test set contains 5000 fake and 5000 real images.

| Category | Generator | Number | Resolution | Real Source |
|---|---|---|---|---|
| Photos | StyleGAN-XL Sauer et al. (2022) | 388 | $256 \times 256$ | ImageNet Deng et al. (2009) |
| | ProGAN Karras (2017) | 424 | $256 \times 256$ | LSUN Yu et al. (2015) |
| | SD v2.1 Rombach et al. (2022) | 361 | $768 \times 768$ | COCO Lin et al. (2014) |
| | SDXL v1.0 Podell et al. (2023) | 380 | $1024 \times 1024$ | COCO Lin et al. (2014) |
| Paintings | StyleGAN 3 Karras et al. (2021) | 623 | $1024 \times 1024$ | WikiArt Tan et al. (2019) |
| | SD v2.1 Rombach et al. (2022) | 131 | $768 \times 768$ | WikiArt Tan et al. (2019) |
| | SDXL v1.0 Podell et al. (2023) | 129 | $1024 \times 1024$ | WikiArt Tan et al. (2019) |
| | Animagine XL | 246 | $1024 \times 1024$ | Danbooru Anonymous et al. (2022) |
| Faces | StyleGAN-XL Sauer et al. (2022) | 509 | $1024 \times 1024$ | FFHQ Karras et al. (2019) |
| | SD v2.1 Rombach et al. (2022) | 295 | $768 \times 768$ | FFHQ Karras et al. (2019) |
| | SDXL v1.0 Podell et al. (2023) | 288 | $1024 \times 1024$ | FFHQ Karras et al. (2019) |
| Other | Midjourney MidJourney (2025) | 626 | $1024 \times 1024$ $1792 \times 1024$ | Photozilla Singhal et al. (2021) |
| | DALL·E 3 Betker et al. (2023) | 600 | $1024 \times 1024$ $1792 \times 1024$ | Photozilla Singhal et al. (2021) |

Table 8: Composition of the complete Chameleon Yan et al. (2024) dataset. After train/test split, the test set contains 1478 fake and 1117 real images. The specific generator used were not reported by the authors.

| Fake Source | Number | Resolution | Real Source | Category |
|---|---|---|---|---|
| Artstation art | 2,976 | | | Scene |
| Civitai civ | 2,016 | various | Unsplash uns | Object |
| Liblib lib | 313 | | | Animal |
| | 5,865 | | | Human |

### 7.3 DETAILED OVERVIEW OF PREVIOUS DETECTORS

Detectors for synthetically generated images leverage a variety of signals, from low-level artifacts to high-level semantic features. Below is a detailed overview of prominent methods and the core ideas behind their approaches.

- **NPR** Tan et al. (2024): This method focuses on *Frequency and Spectral Analysis*. It operates on the principle that up-sampling operations in generative models introduce predictable artifacts in the frequency domain. NPR analyzes an image's frequency spectrum to identify these high-frequency inconsistencies.

- **DIF** Sinitsa & Fried (2024): Also a method based on *Frequency and Spectral Analysis*, DIF aims to extract a "Deep Image Fingerprint." It leverages frequency-aware clues to find unique, model-specific signatures for detection and lineage analysis.

- **PatchCraft** Zhong et al. (2023): This detector is based on *Texture and Patch Analysis*. It posits that artifacts are more pronounced at a local level and works by analyzing the inter-pixel correlation contrast between rich and poor texture regions within an image.

- **SSP** Chen et al. (2024b): Following the *Texture and Patch Analysis* approach, SSP operates on local patches and gradients to identify discriminative features that separate real images from generated ones.

- **DNF** Zhang & Xu (2023): This method uses *Noise Pattern Analysis*. It is designed to extract the unique residual noise patterns present in synthetic images by estimating and analyzing the noise added during the diffusion process.

- **DIRE** Wang et al. (2023): A *Diffusion Process-Based Method* that relies on reconstruction error. It is founded on the principle that diffusion models can reconstruct their own generated images with significantly lower error than they can reconstruct real-world images. This pixel-space error is used as the primary feature for detection.

- **AEROBLADE** Ricker et al. (2024): This method also uses reconstruction error but measures it in the latent space of the model's autoencoder, providing a different perspective on the reconstruction fidelity.

- **LaRE**[2] Luo et al. (2024): This approach uses the latent reconstruction error not as a direct feature, but as a guiding signal for a larger, more complex classification network.

- **DRCT** Chen et al. (2024a): A non-error-based diffusion method that employs contrastive training. It cleverly uses the diffusion model's reconstruction ability to generate hard negative training samples (reconstructed real images labeled as fake) to improve detector robustness.

- **DistilDIRE** Lim et al. (2024): This method addresses the slow speed of error-based detectors by distilling the knowledge from a large, slow DIRE-based detector into a much smaller and faster one, which operates without a full reconstruction cycle.

- **UniFD** Ojha et al. (2023): A pioneering *Pretrained Feature-Based* detector. It leverages the rich, semantic feature space of large vision-language models like CLIP, demonstrating that a simple linear classifier trained on these general-purpose features can achieve impressive generalization across unseen generators.

- **LASTED** Wu et al. (2023): This detector also leverages vision-language models but through the specific mechanism of language-guided contrastive learning to better align features for the detection task.

- **Bi-LoRa** Keita et al. (2025): This approach creatively reframes the detection problem as an image captioning task. It fine-tunes a VLM to output a simple caption of "real" or "fake," leveraging the model's generative language capabilities for classification.

- **AIDE** Yan et al. (2024): This detector proposes a *Hybrid Model* that combines the best of both worlds. It fuses high-level semantic features from a pretrained CLIP model with specialized, hand-crafted modules (like DCT and SRM filters) designed to capture low-level, artifact-based texture statistics.

## 7.4 ADDITIONAL EXPERIMENT RESULTS

Here we included the supplementary experiment results on our evaluation sets and report AP and accuracy of each class: Real & Fake.

Table 9: Comparison of detectors on established benchmarks based on real and fake class accuracy. Detectors marked with * were evaluated using their official pretrained weights. Results are in the format: rAcc / fAcc (%).

| Detector | GenImage | ImagiNet | Chameleon | STD | Mean |
|---|---|---|---|---|---|
| BiLoRA* | 99.71 / 22.75 | 40.76 / 72.36 | 62.68 / 34.38 | 29.80 / 25.95 | 67.72 / 43.16 |
| DIF* | 96.4 / 87.2 | 99.5 / 1.1 | 86.3 / 20.4 | 6.90 / 45.18 | 94.07 / 36.23 |
| UNIFD* | 97.35 / 43.63 | 86.21 / 67.32 | 97.94 / 3.58 | 6.61 / 32.22 | 93.83 / 38.18 |
| NPR* | 58 / 93.4 | 61.7 / 87.5 | 98.31 / 10.16 | 22.28 / 46.45 | 72.67 / 63.69 |
| DNF | 100 / 55.8 | 99.8 / 49.04 | 100 / 6.26 | 0.12 / 26.86 | 99.93 / 37.03 |
| UNIFD | 93.25 / 54.33 | 89.53 / 58.53 | 75.63 / 53.49 | 9.29 / 2.70 | 86.14 / 55.45 |
| LASTED | 92.21 / 54.55 | 94.49 / 54.12 | 94.99 / 45.93 | 1.48 / 4.86 | 93.90 / 51.53 |
| AIDE | 99.35 / 77.67 | 98.46 / 40.1 | 99.93 / 6.1 | 0.74 / 35.80 | 99.25 / 41.29 |
| LASTED* | 90.83 / 95.13 | 73.18 / 52.5 | 61.25 / 16.12 | 14.88 / 39.55 | 75.09 / 54.58 |
| NPR | 97.9 / 75.0 | 96.9 / 48.6 | 99.9 / 4.8 | 1.53 / 35.46 | 98.23 / 42.80 |
| AIDE* | 99.82 / 74.4 | 88.2 / 57.1 | 94.01 / 18.97 | 5.81 / 28.36 | 94.01 / 50.16 |
| SSP* | 99.3 / 87.88 | 91.56 / 59.66 | 99.53 / 4.21 | 4.54 / 42.57 | 96.80 / 50.58 |
| SSP | 97.25 / 89.43 | 94.98 / 61.0 | 99.8 / 5.85 | 2.41 / 42.50 | 97.34 / 52.09 |
| **Ours** | 96.95 / 68.12 | 91.50 / 72.26 | 94.23 / 51.11 | 2.73 / 11.21 | 94.23 / 63.83 |

Table 10: Comparison of detectors on our proposed **OmniGen** test set based on real and fake class accuracy. Detectors marked with * were evaluated using their official pretrained weights. Results are in the format: rAcc / fAcc (%).

| Generator | DIF* | UNIFD* | UNIFD | DNF | LASTED* | BiLora* | AIDE* | AIDE | SSP* | NPR | LASTED | SSP | Ours |
|---|---|---|---|---|---|---|---|---|---|---|---|---|---|
| GPT-4o | 99.9 | 79.3 | 64.73 | 100 | 87.33 | 63.10 | 94.5 | 94.3 | 99.8 | 100 | 89.34 | 99.6 | 99.10 |
|  | 5.8 | 57.4 | 83.09 | 51.44 | 77.26 | 96.03 | 75.27 | 62.45 | 93.1 | 98.4 | 81.4 | 96.03 | 93.32 |
| Imagen 4 | 99.9 | 97.78 | 74.52 | 100 | 57.89 | 63.10 | 94.5 | 99.9 | 99.8 | 100 | 96.99 | 99.6 | 99.3 |
|  | 6.5 | 1.41 | 49.55 | 45.59 | 87.88 | 95.79 | 3.9 | 2.3 | 13.63 | 77.2 | 96.39 | 50.10 | 95.49 |
| Imagen 4 Ultra | 99.9 | 97.8 | 74.57 | 100 | 44.22 | 63.10 | 94.50 | 99.9 | 99.6 | 100 | 97.36 | 100 | 99.30 |
|  | 6.2 | 2.1 | 49.15 | 45.3 | 92.8 | 96.8 | 5.1 | 2.5 | 14.9 | 74.6 | 96.30 | 51.2 | 93.40 |
| FLUX 1 dev | 99.9 | 97.8 | 64.04 | 100 | 52.0 | 63.10 | 94.5 | 94.3 | 99.6 | 100 | 91.44 | 99.5 | 99.10 |
|  | 15.3 | 0.7 | 76.88 | 73.4 | 92.2 | 97.90 | 72.3 | 73.0 | 96.0 | 88.4 | 85.40 | 93.80 | 97.80 |
| Kandinsky 3 | 99.9 | 73.4 | 64.06 | 100 | 69.56 | 63.10 | 94.5 | 94.3 | 99.80 | 100 | 95.83 | 99.8 | 99.10 |
|  | 5.02 | 49.4 | 71.27 | 75.65 | 96.90 | 77.30 | 68.3 | 85.8 | 95.7 | 85.7 | 87.0 | 95.50 | 99.5 |
| PixArt-δ | 99.9 | 74.6 | 64.06 | 100 | 80.82 | 63.10 | 94.5 | 94.3 | 99.60 | 100 | 95.67 | 99.8 | 99.10 |
|  | 2.7 | 45.5 | 91.39 | 77.4 | 84.30 | 86.20 | 76.2 | 82.3 | 93.5 | 68.6 | 85.30 | 89.8 | 98.90 |
| Juggernaut v11 | 99.9 | 57.9 | 64.06 | 100 | 87.22 | 63.10 | 94.5 | 94.3 | 99.70 | 100 | 96.01 | 99.6 | 99.10 |
|  | 3.25 | 47.9 | 87.89 | 19.5 | 95.0 | 90.40 | 91.8 | 93.7 | 93.5 | 82.6 | 91.89 | 94.8 | 99.20 |
| Dreamshaper | 99.9 | 35.7 | 64.06 | 100 | 81.53 | 63.10 | 94.5 | 94.3 | 99.8 | 100 | 96.15 | 99.9 | 99.10 |
|  | 6.3 | 73.0 | 90.49 | 30.4 | 89.0 | 86.40 | 96.6 | 96.1 | 99.0 | 66.2 | 97.6 | 99.85 | 97.70 |
| CogView4-6B | 99.9 | 97.8 | 64.06 | 100 | 46.32 | 63.10 | 94.5 | 94.3 | 99.6 | 100 | 93.69 | 99.8 | 99.10 |
|  | 1.2 | 2.0 | 97.0 | 22.9 | 66.10 | 85.10 | 94.7 | 95.3 | 98.6 | 95.2 | 87.4 | 85.9 | 100 |
| HiDream-I1 | 99.9 | 97.8 | 64.04 | 100 | 54.02 | 63.10 | 94.5 | 94.3 | 99.8 | 100 | 92.22 | 99.8 | 99.10 |
|  | 1.8 | 1.1 | 72.87 | 79.2 | 95.30 | 86.30 | 28.9 | 86.1 | 87.6 | 75.4 | 91.0 | 94.0 | 96.60 |
| SD3.5-medium | 99.9 | 57.6 | 63.92 | 100 | 83.96 | 63.10 | 94.5 | 94.3 | 99.4 | 100 | 95.56 | 84.5 | 99.10 |
|  | 3.1 | 55.6 | 89.35 | 45.3 | 66.20 | 96.60 | 65.4 | 80.6 | 88.7 | 90.9 | 78.4 | 99.8 | 97.30 |
| MidJourney v7 | 99.9 | 73.5 | 63.92 | 100 | 72.19 | 63.10 | 94.5 | 94.3 | 99.7 | 100 | 95.44 | 99.9 | 99.10 |
|  | 4.3 | 62.6 | 52.46 | 41.94 | 49.35 | 96.40 | 58.45 | 89.28 | 73.47 | 49.2 | 88.09 | 96.0 | 77.8 |
| STD | 0.00 | 20.50 | 4.07 | 0.00 | 16.39 | 0.00 | 0.00 | 2.18 | 0.13 | 0.00 | 2.45 | 4.41 | 0.08 |
|  | 3.68 | 28.92 | 17.23 | 23.37 | 14.96 | 6.60 | 32.20 | 33.34 | 30.97 | 13.88 | 6.02 | 17.51 | 6.02 |
| Mean | 99.90 | 78.42 | 65.84 | 100.00 | 68.09 | 63.10 | 94.50 | 95.23 | 99.68 | 100.00 | 94.64 | 98.48 | 99.13 |
|  | 5.12 | 33.22 | 75.95 | 50.67 | 82.69 | 90.94 | 61.41 | 70.79 | 78.98 | 79.37 | 88.85 | 87.23 | 95.58 |

Table 11: Comparison of detectors on our proposed OmniGen test set on Average Precision (%). Models marked with * were evaluated using their official pretrained weights. Best results are in **bold** and second best is underlined.

| Generator | UNIFD* | BILORA* | LASTED* | UNIFD | AIDE* | LASTED | AIDE | SSP* | SSP | DNF | NPR | DIF* | Ours |
|---|---|---|---|---|---|---|---|---|---|---|---|---|---|
| GPT-4o | 58.7 | 58.1 | 72.0 | 81.2 | 90.4 | 75.9 | 85.3 | 99.8 | 99.9 | 99.7 | **100** | 99.7 | 99.7 |
| Imagen 4 | 46.1 | 71.2 | 66.8 | 65.9 | 50.8 | 93.8 | 53.3 | 87.2 | 92.5 | **100** | 99.8 | 100 | 99.8 |
| Imagen 4 Ultra | 46.7 | 71.7 | 56.6 | 64.8 | 50.9 | 93.6 | 53.2 | 86.9 | 92.7 | **100** | 99.6 | 100 | 99.8 |
| FLUX 1 dev | 39.4 | 72.2 | 68.1 | 73.4 | 94.2 | 79.2 | 93.9 | 99.8 | 99.8 | **100** | 99.9 | 99.4 | 99.9 |
| Kandinsky 3 | 63.4 | 63.7 | 79.7 | 70.7 | 93.2 | 89.2 | 96.9 | 99.8 | 99.8 | 100 | 99.9 | 100 | **100** |
| PixArt-δ | 60.4 | 67.3 | 74.0 | 84.3 | 93.7 | 87.9 | 96.0 | 99.0 | 99.0 | 100 | 99.8 | 99.9 | **100** |
| Juggernaut v11 | 51.1 | 69.0 | 81.8 | 83.1 | 98.1 | 89.7 | 98.7 | 99.5 | 99.8 | 97.8 | 99.7 | 99.9 | 99.2 |
| Dreamshaper | 52.5 | 67.3 | 75.7 | 84.0 | 99.0 | 96.7 | 99.1 | 99.9 | **100** | 98.9 | 99.7 | 100 | 99.9 |
| CogView4-6B | 48.5 | 66.8 | 72.8 | 91.5 | 98.8 | 84.6 | 99.0 | 99.8 | 98.8 | 97.8 | 100 | 100 | **100** |
| HiDream-I1 | 37.6 | 67.3 | 70.6 | 71.3 | 71.4 | 84.5 | 96.8 | 98.3 | 99.3 | **100** | 99.8 | 100 | 99.9 |
| SD3.5-medium | 55.0 | 71.6 | 72.2 | 85.6 | 91.6 | 78.3 | 96.2 | 99.3 | 99.2 | 100 | 99.9 | **100** | 99.9 |
| MidJourney v7 | 73.2 | 71.5 | 58.6 | 61.3 | 88.1 | 86.8 | 97.7 | 97.6 | 99.8 | 99.8 | 98.9 | **99.98** | 98.3 |
| STD | 10.18 | 4.12 | 7.48 | 9.72 | 17.55 | 6.50 | 17.02 | 4.82 | 2.72 | 0.90 | 0.30 | **0.18** | 0.51 |
| Mean | 52.72 | 68.14 | 70.74 | 76.42 | 85.03 | 86.68 | 88.83 | 97.24 | 98.38 | 99.49 | 99.77 | **99.91** | 99.69 |

