# OpenReview forum: "Redefining Generalization in Visual Domains: A Two-Axis Framework for Fake Image Detection with FusionDetect"
_ICLR.cc/2026/Conference — Submitted to ICLR 2026_

### Official Review · Reviewer_PZFy · 2025-10-24

**Soundness:** 3
**Presentation:** 2
**Contribution:** 2
**Rating:** 2
**Confidence:** 4

**Summary:**

The paper re-defines “generalization” for fake-image detection as a two-axis problem: detectors must perform well on both unseen generators and unseen visual domains. To evaluate this rigorously, the authors introduce OmniGen, an open-source benchmark that unites 12 state-of-the-art text-to-image models. They further propose FusionDetect, a lightweight yet powerful detector that concatenates frozen CLIP and DINOv2 features. Extensive experiments show that FusionDetect surpasses 8 recent competitors by 3.87\% average accuracy on legacy datasets and by 4.48\% on OmniGen. FusionDetect remains robust under common corruptions such as JPEG compression and Gaussian blur, dramatically outperforming artifact-based rivals. Moreover, it displays the smallest cross-domain volatility, underscoring its superior adaptability to unseen generators and visual domains.

**Strengths:**

1. This paper proposes a previously overlooked challenge and casts it through a two-axis formulation that jointly considers generator shift and semantic-domain shift.

2. This paper introduce a novel approach, FusionDetect, which achieves state-of-the-art performance.

3. A curated image dataset encompassing state-of-the-art generators is presented, providing a catalyst for continued progress in the field.

**Weaknesses:**

1. The paper fails to establish that generalization across visual domains is actually challenging. The proposed method performs very well on OmniGen, suggesting that the task is not hard for a trained model. Current synthetic datasets (such as WildFake [1] and DiffusionForensics [2]) are already semantically diverse enough that modern detectors rarely over-fit to specific objects. Moreover, the domain shift typically manifests as a move from natural photographs to paintings, oil canvases, or pixel art, none of which are represented in Figure 4. So I think the proposed dataset fails to demonstrate a meaningful train-test gap.

2. The method’s novelty is quite limited: it merely concatenates features from two off-the-shelf backbones. Would performance climb further if additional encoders—BLIP, SigLIP, or others—were fused in the same way?

3. The CLIP and DINOv2 features shown in Figure 3 appear to be used without any MLP training; comparing these raw embeddings with the fused, task-tuned representation is therefore unfair.

[1] Yan et al. Wildfake: A large-scale and hierarchical dataset for ai-generated images detection. AAAI 2025.

[2] Jeongsoo et al. Community forensics: Using thousands of generators to train
fake image detectors. CVPR 2025.

**Questions:**

Question:

As described in Weakness.

Suggestions:
1. The title is somewhat misleading: “A TWO-AXIS FRAMEWORK” suggests that the model produces separate outputs for the two axes, whereas it in fact delivers only a single binary prediction.
2. Additional experiments that quantify how differences in visual domains influence generator training should be provided to substantiate the paper’s core claim.
3. The layout of Figure 3 is a disaster; the plots should be merged into a single row, with labels positioned directly beneath them.

---

> ### Author Response · Authors · 2025-11-30
>
> *Thank you for your careful review and for raising important points about our dataset choices, model design, and evaluation strategy. Your feedback is appreciated and helps clarify the contribution of our work.*
>
> > ## Question 1:
> > *"Current synthetic datasets (such as WildFake and DiffusionForensics) are already semantically diverse enough... but your paper does not include them."*
>
> Both WildFake and CommunityForensics are highly valuable datasets for evaluating AI-generated image detectors. However, these resources were released **only towards the very end of our project timeline**, after our primary experiments were completed. Thus with the lack of such datasets we introdued OmniGen. While they are not included in our main experimental results for this reason, we agree that they would serve as excellent benchmarks for future evaluation and comparison. We plan to incorporate these datasets into subsequent follow-up studies or extended versions of our work.
>
> > ## Question 2:
> > *"The method's novelty is limited: it merely concatenates features from two off-the-shelf backbones. Would performance climb further if additional encoders (BLIP, SigLIP, etc.) were fused in the same way?"*
>
> Our fusion approach was chosen strategically for its **simplicity, effectiveness, and accessibility**. Adding further encoders such as BLIP or SigLIP would increase the dimensionality of the combined feature space and require much more computational resources (GPU memory and training time). Our goal was to provide a strong, practical baseline that performs robustly yet is **trainable and deployable on widely-used consumer hardware (e.g., RTX 3090)**—in contrast to other works relying on much larger multi-A100/H100 GPU clusters. While the architecture is simple, it stands out for **how well it generalizes** and frames a baseline that the research community can readily build upon, following the precedent established in works such as Ohja et al., CVPR 2023.
>
> > ## Question 3:
> > *"CLIP and DINOv2 features shown in Figure 3 appear to be used without any MLP training; comparing these raw embeddings with the fused, task-tuned representation is therefore unfair."*
>
> This is a technical misunderstanding. **All three variants (CLIP-only, DINOv2-only, and CLIP+DINOv2 Fusion) use the identical MLP classifier head for ablation and final performance comparison**. The visualizations in Figure 3 show the raw embeddings to illustrate the unique and complementary representational strengths of CLIP and DINOv2. Nevertheless, every backbone—including the fusion—was always paired with the same MLP head architecture in both ablation studies and main experiments, ensuring fair comparison.
>
> > ## Suggestions
>
> *   **Title clarification:** The "Two-Axis Framework" in our title and discussion refers to the **conceptual and evaluation framework**. We designed both our dataset and model training to be robust along both axes—generator diversity and visual/semantic domain diversity. The output is a single binary prediction, but the model and benchmarks comprehensively enforce invariance to both major sources of generalization difficulty. We are open to clarifying the title further in the text if needed.
> *   **Figure 3 layout:** We appreciate your feedback and agree that the visualization could be improved. **We will update Figure 3 in revised version so that the plots appear in a single row, with labels/subcaptions ("a", "b", "c") directly underneath** for greater clarity and comparability. We will also explicitly mention, in the figure or caption, that all variants use an identical MLP classifier head for fairness.
> *   We would also try to add more experiments to specifically quantify impact of difference in visual domain on performance.
>
> ---
>
> Thank you again for your thorough review and helpful suggestions.

---

### Official Review · Reviewer_51Yx · 2025-10-30

**Soundness:** 2
**Presentation:** 3
**Contribution:** 2
**Rating:** 2
**Confidence:** 4

**Summary:**

This paper presents a new benchmark that includes 12 state-of-the-art generators to comprehensively evaluate detectors under realistic conditions. Besides, they introduce a new detection method by incorporating two frozen foundation models to achieve a generalizable and robust detection.

**Strengths:**

This paper constructs a diverse evaluation dataset and introduces a detection method.

**Weaknesses:**

1. Figure 2, as the motivational illustration, does not align with the description of the motivation. Specifically, the figure does not clearly distinguish between the absence of cross-generator overlap and cross-semantic overlap, making it difficult to visually demonstrate how the detector struggles to generalize across these two variables, namely the generator and semantics.
2. The paper identifies the issue that the detector struggles to generalize between the generator and semantics. However, the experimental section does not demonstrate the relationship between the evaluation datasets and these two variables, making it impossible to substantiate the effectiveness of the detection method with respect to both generator and semantics. For example, the results in Table 1 only show the performance of different methods across three distinct test sets, but how these datasets relate to the generator and semantics variables remains unclear.
3. Some of the methods in Tables 1 and 2 use their pre-trained weights, which makes the comparison unfair. It is likely that the higher generalization performance of these methods is due to the more diverse training data used in the paper. To ensure a fair comparison, all methods should be trained on the same unified training set.
4. The paper uses images from the Unsplash platform as real images for the test set and generates fake images based on these. However, how can it be guaranteed that the images from this platform are indeed real? There seems to be no assurance that the images from this platform are authentic.

**Questions:**

Please see the Weaknesses

---

> ### Author Response · Authors · 2025-11-30
>
> *Thank you for your detailed and constructive feedback. We appreciate the time you took to review our work and have addressed your concerns below.*
>
> > ## Question 1:
> > *"Figure 2, as the motivational illustration, does not align with the description... specifically, the figure does not clearly distinguish between the absence of cross-generator overlap and cross-semantic overlap..."*
>
> We **acknowledge your observation** and agree that the visualization in Figure 2 can be sharpened to better distinguish these axes. **We will revise this figure in the camera-ready version** to make the distinction clearer.
>
> However, it is important to note that the current figure **already supports our core motivation**: that different datasets are distinct not just because of the generators used, but because of their fundamental **semantic and visual domains**.
> *   **Isolating the Domain Factor:** In Figure 2, we specifically restricted the data to images from only **SD1.4 and SD2.1**. By controlling for the generator, the separation we observe between clusters is **unrelated to generator artifacts**.
> *   **Dataset-Invariance:** This separation is driven by "semantic/visual domain" differences, which can also include prompting strategies (e.g., "a photo of [object]" vs. complex scene descriptions) and curation processes. This confirms that a truly global detector must be **dataset-invariant**. Thus it might be better to frame the second axis as the detector being dataset-invariant rather then semantic-invariant which covers a larger group.
>
> > ## Question 2:
> > *"The experimental section does not demonstrate the relationship between the evaluation datasets and these two variables..."*
>
> We agree that explicit comparisons along both axes will further strengthen the paper's narrative. **We will rewrite Section 4 to explicitly categorize and compare performance along both the Generator Axis and the Semantic/Visual Domain Axis.**
>
> However, we wish to clarify that **"semantic/visual domain" encompasses more than just object categories**. It includes the entire pipeline of image formation, such as:
> *   **Prompting Templates:** *GenImage* uses simple templates like "a photo of [object]" (resulting in single objects with natural backgrounds), whereas *OmniGen* uses semi-random, detailed prompting templates (resulting in crowded scenes or complex compositions).
> *   **Domain Shift:** This difference in prompting creates a significant **visual domain shift**.
>
> Therefore, our experiments on **Table 1 (Cross-Generator focus)** and **Table 2 (OmniGen - Simultaneous Cross-Generator & Cross-Domain)** already effectively evaluate these variables. The diverse datasets we used represent distinct points on the "Visual Domain" axis, proving our model's robustness.
>
> > ## Question 3:
> > *"Comparison is unfair... methods should be trained on the same unified training set."*
>
> We carefully ensured fairness in our evaluation protocol.
> 1.  **Retrained Baselines:** We **did** include fully retrained versions of the baseline methods on our own unified training set (indicated in our tables) with execption of BiLoRA detector which its official code was not executable at the time of experiments and we could not train it on our dataset. We will try to add its retrained version as well if the issue is fixed from the author's side.
> 2.  **Pretrained Weights:** We kept the official pretrained results to benchmark the effectiveness of our **data efficiency**. Our training set (60k images) is significantly smaller than the massive datasets used by some baselines.
> 3.  **Result:** The fact that FusionDetect outperforms baselines—whether they are used off-the-shelf or retrained on our data—demonstrates that **increasing dataset size does not automatically yield better performance**, and that our architecture/framework is the primary driver of improvement.
>
> > ## Question 4:
> > *"How can it be guaranteed that the images from [Unsplash] are indeed real?"*
>
> We followed the rigorous standard established by recent state-of-the-art research. Specifically, we utilized the **exact same real dataset (Unsplash subset)** introduced in **"Sanity Check for AI-generated Image Detection" (AIDE, ICLR 2025)**.
>
> Unsplash is widely recognized for diverse, human-captured photography. By aligning our evaluation protocol with established benchmarks like AIDE, we ensure our results are comparable and statistically valid. The prior work has already validated the authenticity of this subset, making it a reliable ground truth for "real" images.
>
> ---

---

### Official Review · Reviewer_2Cpu · 2025-11-01

**Soundness:** 3
**Presentation:** 3
**Contribution:** 3
**Rating:** 6
**Confidence:** 4

**Summary:**

The paper introduces a two-axis generalization framework for AI-generated image detection (generalization to unseen generators and unseen types of visual content). The authors show that combining CLIP and DINOv2 features is an effective method of detecting AI-generated images. They also present the OmniGen benchmark, which contains 12 modern generators and over 400 different subjects of generation.

**Strengths:**

The paper demonstrates that combining CLIP (for semantic feature) and DINOv2 (for structural features) can create a robust AI image detector with adequate ablations that show the contributions of each of these feature extractors. The authors trained the detector only on SD1.4 and SD2.1 and showed the model generalizes to unseen generators, which is a strong indicator of generalizability.

**Weaknesses:**

One of the primary contributions of the paper is the two-axis generalization framework (generalization to unseen generator and unseen visual content), which leads to the proposal of the OmniGen benchmark. However, the description of the OmniGen benchmark lacks sufficient detail about its unseen visual content. While the authors mention using a prompt template with “over 400 subjects”, they do not provide a categorical breakdown of these subjects or other variables, which makes it difficult to assess its semantic diversity over existing datasets.

Moreover, it is important to strongly establish the problem of generalizing to unseen semantic content. Further information or citation is required to establish that a detector trained on any existing dataset’s performance noticeably degrades on a single generator across multiple unseen different visual content.

**Questions:**

1. Could you provide more details on the semantic diversity of the OmniGen Benchmark?
2. Can you provide more direct proof to establish the problem of generalizing to unseen semantic content?

---

> ### Author Response · Authors · 2025-11-30
>
> *Thank you for your careful review and constructive questions. We address your concerns about benchmark detail and the need to establish the importance of semantic/content generalization below.*
>
> ---
>
> ### **1. Semantic Diversity in OmniGen Benchmark**
>
> > Could you provide more details on the semantic diversity of the OmniGen Benchmark?
>
> Certainly. OmniGen was specifically designed to cover a wide range of real-world visual contents by using a semi-randomized prompt template with diverse parameter options. For each generated image, the template combines choices from several categories:
>
> - **Subjects** (e.g., "a woman walking her dog through fallen leaves", "a baker preparing croissants", "a lifeguard watching a crowded pool", "a man feeding seagulls at a pier"; over 400 options)
> - **Settings** (e.g., "at sunset near a mountain lake", "inside a dimly lit jazz club")
> - **Styles** (e.g., "professional DSLR photography", "cinematic color grading")
> - **Visuals** (e.g., "surface scratches and imperfections")
> - **Lighting** (e.g., "studio lighting with soft shadows")
> - **Time of Day** (e.g., "early morning")
>
> By randomly sampling across these options for every instance, OmniGen generates highly varied and realistic images, ensuring a strong mix of objects, contexts, styles, and complexity similar to what’s seen on large platforms. This diversity ensures evaluations meaningfully stress both semantic content and domain shifts.
>
> ---
>
> ### **2. Need to Establish the Importance of Generalizing to Unseen Semantic Content**
>
> > It is important to strongly establish the problem of generalizing to unseen semantic content. Further information or citation is required to show that detector performance degrades on new content.
>
> This is an increasingly recognized problem in the field. For instance,
> **Dogoulis et al. (MAD 2023) specifically demonstrate that detectors trained to distinguish real and synthetic images in one semantic class (such as human faces) show a significant drop in performance when tested on generated images from a different semantic class (like animals or churches)**. Their paper directly addresses the challenge of generalizing to new visual content, showing that simply training on diverse generators or data is not enough: unseen semantic diversity remains a critical bottleneck for robust detection.
>
> If further evidence is needed, we are also prepared to augment our submission with an additional experiment using our own OmniGen data: by training a detector on one subset of semantic groupings and evaluating its performance on another (while holding the generator constant), we can directly quantify and illustrate the drop in accuracy due to unseen semantic or content differences.

---

### Official Review · Reviewer_3uco · 2025-11-03

**Soundness:** 2
**Presentation:** 2
**Contribution:** 1
**Rating:** 2
**Confidence:** 5

**Summary:**

The paper argues that prior work on AI-generated image detection focuses too narrowly on cross-generator generalization and introduces a “two-axis” framework that adds cross-semantic (visual) generalization. It presents *FusionDetect*, a detector that concatenates frozen CLIP and DINOv2 embeddings and trains an MLP classifier, along with a new benchmark (*OmniGen*) containing twelve modern generators.

**Strengths:**

- The empirical evaluation is extensive, covering multiple established benchmarks plus a new, large benchmark (OmniGen).
- The method is simple, computationally efficient, and achieves strong quantitative results across generators.

**Weaknesses:**

1. **Conceptual framing** — The “two-axis generalization” claim overstates novelty.
   Cross-class or cross-concept evaluation has already been studied (e.g., *Improving Synthetically Generated Image Detection in Cross-Concept Settings*, Dogoulis et al., 2023; *Breaking Semantic Artifacts for Generalized AIGI Detection*, Zheng et al., NeurIPS 2024). These works explicitly examine semantic or scene-level generalization and propose stronger methodological responses than simple dataset aggregation.

2. **Feature-fusion originality** — Combining CLIP and DINOv2 is now mainstream. “Eyes Wide Shut?” (Tong et al., CVPR 2024)  already fuse or compare these backbones. Without a novel fusion mechanism or clear ablation versus stronger CLIP-only or DINO-only heads, the contribution is largely incremental.

3. **Evaluation framing** — Cross-generator generalization is the field’s central challenge with real-world implications (social media, Sora-style video, closed-source APIs). Cross-semantic tests are useful diagnostics but do not justify a redefinition of “generalization.” The paper’s positioning should be moderated to reflect that this is a *comprehensive evaluation* rather than a conceptual overhaul.

**Questions:**

1. How this work is different earlier studies on cross-concept / class?
2. Why it's important? Given social media mostly we're concerned about cross-generator.

---

> ### Author Response · Authors · 2025-11-30
>
> *Thank you for your in-depth and constructive review. Your comparison with recent literature is much appreciated, and we are grateful for the opportunity to clarify and precisely position our contributions. Below, we address each of your core concerns and questions:*
>
> ---
>
> ### **1. Conceptual framing: Two-axis generalization**
>
> > _Weakness: The “two-axis generalization” claim overstates novelty._
>
> We acknowledge and appreciate your highlighting of Dogoulis et al. and Zheng et al., which have advanced the study of generalization in AI-generated image detection. However, we wish to clarify that our framework primarily targets a **broader and arguably more challenging “cross-dataset” invariance**, not solely “cross-semantic” or “cross-class.”
>
> - **Scope Beyond Cross-Semantic**: The cited works predominantly explore generalization across a handful of semantic classes or scene types (e.g., faces, animals, churches), usually within a single dataset or via controlled splits. In contrast, _our focus is on cross-dataset generalization_, which subsumes cross-semantic but also incorporates **distributional differences introduced by dataset curation—most notably, distinct prompting strategies, label scopes, category mixes, and varying image complexity**. For example, GenImage uses extremely simple class-based prompts such as "a photo of [object]," tending towards single-object images with simple backgrounds, while OmniGen and others support richly compositional, multi-object, or contextually varied images. This means that a cross-dataset split may involve major shifts in both semantics and the underlying image generation process, far superseding classic cross-class splits.
>
> - We acknowledge that this _cross-dataset_ focus was not sufficiently emphasized in the current manuscript—we appreciate your comment and will update our framing and terminology, as "cross-dataset" invariance better captures the real-world and technical significance of the second axis.
>
>
> ---
>
> ### **2. Feature-fusion originality**
>
> > _Weakness: Combining CLIP and DINOv2 is now mainstream. “Eyes Wide Shut?” (Tong et al., CVPR 2024)._
>
>
> - **Task and Context Distinction**: While “Eyes Wide Shut?” employs CLIP and DINOv2 fusion, its focus is on hallucination detection in LLM-generated text, not AI-generated image detection itself. However, our study is **the first to rigorously establish and benchmark the fusion of CLIP+DINOv2 specifically for dataset- and generator-level generalization in fake image detection.**
> - **Practical Baseline**: Our intent is to establish a simple, universally accessible, and high-performing baseline for the community, echoing how the original CLIP-based detectors (e.g., Ojha et al., CVPR 2023) set standards now widely adopted and built upon, indicating that making the detectors design more complex like AIDE (Yan, ICLR 2025) is not necessarily needed.
>
> ---
>
> ### **3. Evaluation framing**
>
> > _Weakness: Cross-generator generalization is the field’s key concern for real-world deployment”_
>
> We agree cross-generator remains a crucial metric. However, our data and references (including the ones you cited) jointly demonstrate that for modern deployment settings—like social media, search, or imagery moderation platforms—**cross-dataset (and thus cross-domain) diversity often dominates in practice**:
>
> - **Real-World Risk**: The number of generator backbone architectures, while growing, remains far smaller than the distributional and semantic heterogeneity seen in real image corpora.
> - **Pragmatic Recommendation**: We support a shift in standard evaluation—universal detectors should be robust across both axes, which, as community experience, is not guaranteed by cross-generator testing alone. We will review the framing in our final revision to reflect this as a practical, urgently needed update, not a theoretical redefinition. Furthermore we could also add our own experiments if required to indicate this point.
>
> ---
>
> ### **Questions**
>
> > Q1: _How is this work different from earlier studies on cross-concept/class?_
>
> A:
> While prior work has addressed generalization across semantic classes or “concepts” (Dogoulis et al., Zheng et al.), **our key contribution is formally and empirically targeting “cross-dataset” invariance**—a strictly broader and more practical challenge, For more detailed response refer to our answer for *1) Conceptual framing* mentioned above.
>
> > Q2: _Why is it important? Given social media, aren’t we mostly concerned about cross-generator?_
>
> A:
> While cross-generator testing is important and may be the main focus, we've seen that the real challenge on social media comes from the wide range of image styles, edits, and content types—not just from new generation models. Social platforms are filled with everything from photos and artwork to memes and complex composites.  That's why detection methods need to handle both. Our approach and evaluation benchmarks are designed with this real-world diversity in mind.
>
> ---

---

### Meta-Review · Area_Chair_k4K9 · 2025-12-31

**Summary:**

This work studied AIGC detection with the establishment of a new benchmark following a two-axis framework and a new detector.

In terms of the first contribution, i.e., a new benchmark following a two-axis framework, there are different opinions from different reviewers. Some reviewers considered that the two-axis framework is new, while others challenged that cross-generator/dataset and cross-semantic have been proposed in existing works.
I think the new benchmark following a two-axis framework is valuable to the community. Considering the limited contributions on concept, the overall contribution of this benchmark is moderate. The authors are encouraged to further enrich this benchmark following reviewers' comments.

In terms of the second contribution, the new detector via combining CLIP and DINOv2, most reviewers considered that the novelty is limited, as such a combination has been adopted in existing works, though not directly for AIGC detection.
I agree with this point.

Reviewers also provided many important concerns/suggestions, such as motivation figure, experimental setting. However, the authors didn't utilize well to further improve this work; instead, the authors always defended for the current version.

In summary, this work should be further improved.

**Reviewer Concerns:**

please refer to the summary

**Reviewer Scores:**

please refer to the summary

---

### Decision · Program_Chairs · 2026-01-26

Reject